# GAMformer: Bridging Tabular Foundation Models and Interpretable Machine Learning

## Abstract

While interpretability is crucial for machine learning applications in safety-critical domains and for regulatory compliance, existing tabular foundation models like TabPFN lack transparency. Generalized Additive Models (GAMs) provide the needed interpretability through their additive structure, but traditional GAM methods rely on iterative learning algorithms (such as splines, boosted trees, or neural networks) that are fundamentally incompatible with the in-context learning paradigm of foundation models. In this paper, we introduce GAMformer, the first tabular foundation model for GAMs that bridges the gap between the power of foundation models and the interpretability requirements of critical real-world applications. GAMformer estimates GAM shape functions in a single forward pass using in-context learning, representing a significant departure from conventional iterative approaches. Building on previous research on tabular foundation models, we train GAMformer exclusively on synthetically generated tables to prevent data leakage. Our experiments demonstrate that GAMformer performs comparably to other leading GAMs across various classification benchmarks.

## 1 Introduction

The importance of interpretability in machine learning is evident, especially in areas where transparency, fairness, and accountability are critical (Barocas & Selbst, 2016; Rudin et al., 2022). Interpretable models are essential for building trust between humans and AI systems by allowing users to understand the reasoning behind the model's predictions and decisions (Ribeiro et al., 2016). This is crucial in safety-critical fields like healthcare, where incorrect or biased decisions can have severe consequences (Caruana et al., 2015). Additionally, interpretability is vital for regulatory compliance in sectors like finance and hiring, where explaining and justifying model outcomes is necessary (Arun et al., 2016; Dattner et al., 2019). Interpretable models also help detect and mitigate bias by revealing the factors influencing predictions, ensuring fair and unbiased decisions across different population groups (Mehrabi et al., 2021).

Tabular foundation models have emerged as powerful tools for tabular data prediction, with notable examples including TabPFN (Hollmann et al., 2025a; 2023; Müller et al., 2022), TabFlex (Zeng et al., 2025), TabICL (Qu et al., 2025), and Mitra (Zhang et al., 2025). Tabular foundation models enable both faster and more accurate predictions on tabular data by predicting entire columns of interest in a single forward pass, while amortizing the inference cost through pretraining on a mix of synthetic and real-world data. However, these powerful tabular foundation models operate as black boxes, lacking the interpretability crucial for safety-critical applications where understanding model decisions is not just preferred but required.

We introduce GAMformer (see Figure 1), the first tabular foundation model for Generalized Additive Models (GAMs). GAMformer addresses the interpretability gap in tabular foundation models by combining the power of in-context learning with the interpretability of GAMs, estimating interpretable shape functions using ICL in a single forward pass. Unlike traditional GAMs that use splines with backfitting algorithms (Hastie & Tibshirani, 1987), Explainable Boosting Machines that employ decision trees and cyclic gradient boosting (Lou et al., 2012; 2013; Caruana et al., 2015), or Neural Additive Models that use multilayer perceptrons (Agarwal et al., 2021), GAMformer distinguishes itself by using a non-parametric, binned representation

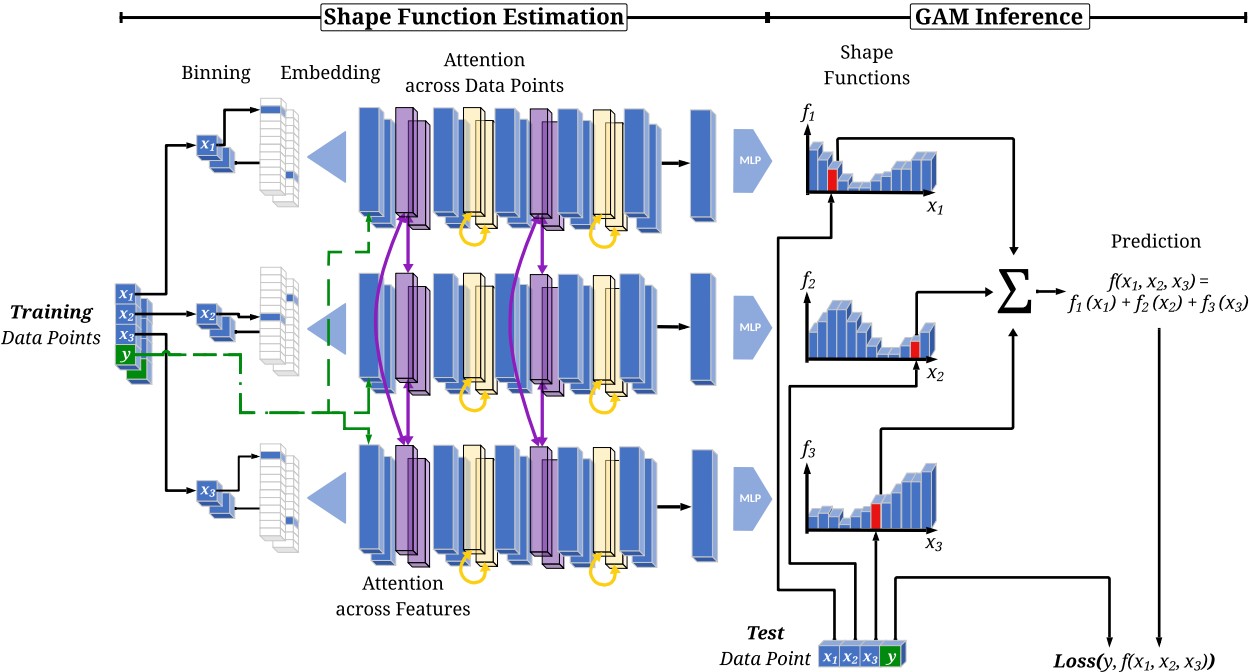

Figure 1: GAMformer's forward pass on a new dataset with three features ($x_1$, $x_2$, $x_3$) and label $y$ and two training and one test-data point: (1) For each training data point, we bin all features, one-hot encode them, embed the resulting vectors and add the label of the data point. (2) We alternate between applying attention across the features and the data points, allowing us to handle varying numbers of each. (3) We decode per-feature shape functions using a shared MLP decoder. (4) We infer the prediction for test data points by looking up and adding each feature's shape function value (red bins) forming the GAM prediction. (5) Finally, we compute the loss based on the prediction allowing the end-to-end training of the shape function estimation based on (in our case, *synthetic*) training datasets.

of shape functions, enabling to model sudden jumps in the shape function occurring for example due to treatment effects in medical data (Caruana et al., 2015). Similar to TabPFN, our model is trained exclusively on large-scale synthetic datasets, yet demonstrates robust performance on real-world data while maintaining full interpretability through its additive structure. Code to reproduce our experiments is available here: `https://anonymous.4open.science/r/gamformer_tmlr-7617/README.md`.

Our main contributions are:

1. We introduce GAMformer, the first tabular foundation model for GAMs which differentiably estimates interpretable shape functions in a single forward pass and enables end-to-end training.

2. Our case study on MIMIC-II demonstrates how GAMformer can be applied to real-world data to generate interpretable models and insights of that data. While our focus is on classification, we demonstrate how the GAMformer architecture can similarly be applied to regression problems.

3. Our experimental results demonstrate GAMformer's capacity to match the accuracy of leading GAMs on various classification benchmarks and by incorporating pairwise posthoc effects allows GAMformer to perform on par with XGBoost (Chen & Guestrin, 2016).

## 2 Background and Related Work

### 2.1 Generalized Additive Models

Generalized Additive Models (GAMs) (Hastie & Tibshirani, 1987) emerged as a generalization of Generalized Linear Models (Nelder & Wedderburn, 1972) which include non-linear transformations of the input features.

The structure of a GAM is given by: equation $g(\mathbb{E}[y|x]) = \beta + \sum_{i=1}^{p} f_i(x_i)$, where $x = (x_1, \ldots x_p) \in \mathcal{X} \subseteq \mathbb{R}^p$ is the input with $p$ features, $y \in \mathcal{Y} \subseteq \mathbb{R}^m$ is the response variable , and $f_i : \mathbb{R} \to \mathbb{R}$ are univariate functions termed *shape functions* that capture the individual contributions of each feature. The intercept $\beta \in \mathbb{R}$ is a learnable bias term, and $g : \mathbb{R} \to \mathbb{R}$ is the link function that connects the expected outcome to the linear predictor, examples of which include the logit or softmax function for binary or multiclass classification or the identity function for regression. The shape functions $f_i$ in GAMs, also sometimes called partial dependence plots, allow for an interpretable representation of each feature's effect, akin to the role of coefficients in linear regression, thus enabling practitioners to inspect the learned potentially non-linear relationships.

Traditional GAMs often use splines and backfitting (Hastie & Tibshirani, 1987), enhanced by penalized regression splines (Wood, 2003) and fast fitting algorithms (Wood, 2001). Spline-based GAMs use the backfitting algorithm, iteratively updating each shape function to fit the residuals of others until convergence. More recent advances include Explainable Boosting Machines (EBMs) (Lou et al., 2012; 2013; Caruana et al., 2015), which use decision trees to model shape functions via cyclic gradient boosting. This approach learns each feature's contribution iteratively in a round-robin manner, mitigating collinearity effects and accurately modeling steps in the data, which is crucial for capturing discontinuities like treatment effects in medical data. On the other hand, Neural Additive Models (NAMs) (Agarwal et al., 2021) and follow up works (Chang et al., 2021; Dubey et al., 2022; Radenovic et al., 2022; Xu et al., 2022; Enouen & Liu, 2022; Bouchiat et al., 2024) use multilayer perceptrons (MLPs) as non-linear transformations to model the shape functions $f_i$. As a result, NAMs can be optimized using variants of gradient descent by leveraging automatic differentiation frameworks. Finally, GAMs have also found applications in time-series forecasting, with models such as Prophet (Taylor & Letham, 2018) and NeuralProphet (Triebe et al., 2021). See Appendix A for an extended related work section.

## 2.2 In-Context Learning & Prior-Data Fitted Networks

In-Context Learning (ICL) was first demonstrated alongside the introduction of GPT-3 (Brown et al., 2020), where the authors showed that Transformer models (Vaswani et al., 2017) could learn to perform tasks solely from input examples, without explicit training or fine-tuning, after self-supervised pre-training. This capability marks a significant paradigm shift from the traditional machine learning paradigm of in-weight learning, where the parameters of a model are adjusted in order to learn a new task. The discovery of ICL has led to numerous investigations into the mechanisms used by trained transformers that enable ICL. Olsson et al. (2022) found that a two-layer attention-only network can develop "induction heads", a mechanism that outputs the token succeeding a previous instance of the current token, precisely when its ICL performance increases. Chan et al. (2022) investigated the properties of the data that contribute to the emergence of ICL abilities, while Reddy (2024) identified factors responsible for the abrupt emergence of induction heads.

Of particular relevance to this paper are Prior-Data-Fitted Networks (PFNs) (Müller et al., 2022; Hollmann et al., 2023), which showed that a transformer trained on complex synthetic data generated using random causal graphs can be used for tabular classification. From a Bayesian perspective, such causal graphs $\phi$ sampled from a hypothesis space $\Phi$ (the prior), define a mechanism that describes the relationship between the input and output variables. In TabPFNs (Hollmann et al., 2023), a synthetic dataset $D \sim p(D) = \mathbb{E}_{\phi \sim p(\phi)}[p(D|\phi)]$ is repeatedly constructed by propagating samples $x \sim p(\mathcal{X})$ from the input space through a randomly sampled structural causal model (SCM), $\phi \sim p(\phi)$, to obtain the corresponding $y$ values. We denote the dataset containing $N$ such examples as the set $D := \{(x^{(n)}, y^{(n)})\}_{n=1}^{N}$. To simulate practical inference scenarios, the dataset $D$ is split into $D_{\text{train}}$ and the context dataset $D_{\text{test}} = D \backslash D_{\text{train}}$. The transformer model parses the pairs $(x_{\text{train}}, y_{\text{train}}) \in D_{\text{train}}$, as well as $x_{\text{test}}$, as single input tokens and its parameters $\theta$ are updated to minimize the negative log likelihood on the test held-out examples: $\mathbb{E}_{(D_{\text{train}} \cup (x_{\text{test}}, y_{\text{test}})) \sim p(D)}[-\log q_\theta(y_{\text{test}}|x_{\text{test}}, D_{\text{train}})]$. Müller et al. (2022) showed that by minimizing this loss, TabPFN approximates the true posterior predictive distribution

$$p(y_{\text{test}}|x_{\text{test}}, D_{\text{train}}) = \int_\Phi p(y_{\text{test}}|x_{\text{test}}, \phi)p(\phi|D_{\text{train}})d\phi \propto \int_\Phi p(y_{\text{test}}|x_{\text{test}}, \phi)p(D_{\text{train}}|\phi)p(\phi)d\phi \qquad (1)$$

on a new input point from the test set $x_{\text{test}}$ up to an additive constant. This paradigm has since been extended to time-series forecasting (Dooley et al., 2024; Bhethanabhotla et al.), hyperparameter optimization

(Müller et al., 2023; Adriaensen et al., 2024; Rakotoarison et al., 2024) and the prediction of neural network weights (Mueller et al., 2025). Similarly, Conditional Neural Processes (Garnelo et al., 2018) also perform a form of ICL, using a neural architecture with weights meta-learned on real data. (Nguyen & Grover, 2022) extended Neural Processes to a transformer architecture, leading to an architecture similar to PFNs. *GAMformer* builds on top of TabPFN by training a transformer on synthetically generated datasets to estimate the shape function per feature and computing predictions by adding the individual shape function values.

## 3 GAMformer

We first provide a high-level overview of how GAMformer works before delving into the details of each of its components. GAMformer follows a differentiable two-step approach, as illustrated in Figure 1. First, a transformer estimates shape functions using ICL on the training dataset $D_{\text{train}}$. Next, using the shape function, we make predictions for each test data point $x_{\text{test}}$. This methodology replaces the traditional data fitting process of GAM variants with a single forward pass of a pre-trained transformer model, eliminating the need for optimization and regularization hyperparameters. We now describe each model component in more detail.

### 3.1 Shape Estimation and Predictions

We obtain the shape functions with ICL by applying a transformer on the training input points and labels: $\tilde{f} = \mathcal{T}_\theta(x_{\text{train}}, y_{\text{train}}) \in \mathbb{R}^{p \times n_{\text{bins}} \times m}$, where $p, m$, and $n_{\text{bins}}$ are respectively the numbers of features, prediction classes and bins. To get predictions on a new test point $x_{\text{test}}$, we first bin each feature value and then apply the estimated shape function: $g(\tilde{y}_{\text{test}}) = \sum_{i=1}^{p} \tilde{f}_{ij_{x_i}} \in \mathbb{R}^m$, where $j_{x_i} \in [n_{\text{bins}}]$ denotes the bin index corresponding to the $i-$th feature of $x_{\text{test}}$. We now give more details on the binning and the architecture used for $\mathcal{T}_\theta$ before discussing the pre-training.

### 3.2 Model Architecture

**Feature Preprocessing.** All features of each data point are binned, one-hot encoded, and finally embedded using an MLP. We use $n_{\text{bins}} = 64$ bins for each feature, allocating bins based on the quantiles of the feature in the training dataset. Similarly to TabPFN, we embed the label of each datapoint and add it to the embedding of each feature. Categorical features are equally distributed across the 64 bins according to their ratios.

**Representation of the shape functions.** To accurately represent the shape functions, we chose to predict a discrete representation for each feature by discretizing it into 64 bins. An alternative approach would have been to predict the weights of a Neural Additive Model (NAM), similar to Mothernet (Mueller et al., 2025). However, we decided against this approach to more naturally represent sudden discontinuities in the shape functions. We refer to our case study on MIMIC-II in Section 4.2 for an illustration of this effect.

**Transformer Architecture.** Our transformer backbone consists of 12 layers that alternate between two types of self-attention: *column-wise attention*, which models interactions across features within a datapoint, and *row-wise attention*, which models interactions across datapoints for a given feature (Lorch et al., 2022; Hollmann et al., 2025b). This alternating or *bi-attention* design allows the model to simultaneously capture feature dependencies and dataset-level structure. Importantly, it also guarantees permutation equivariance: reordering features or samples only reorders the outputs accordingly, ensuring that the representation is independent of the arbitrary input order. As a result, the backbone can handle datasets with varying numbers of datapoints and features without requiring padding to a fixed size, in contrast to TabPFN v1(Hollmann et al., 2023). This makes the architecture both more flexible and more efficient when applied to diverse tabular learning tasks.

After the transformer layers, we compute the average embeddings for each class based on training labels enabling multi-class classification (limited to 10 classes in our experiments). This averaging yields one embedding per class per feature which we denote $h \in \mathbb{R}^{p \times d \times m}$ where $d$ denotes the embedding dimension of

the transformer[1]. Each embedding is then passed through a shared decoder MLP to produce the binned shape functions $\tilde{f} \in \mathbb{R}^{p \times n_{\text{bins}} \times m}$, allowing sharing of parameters across features and classes. The model comprises 40k parameters in the encoder layer, 50.5M parameters in the transformer layers, and 0.3M parameters in the decoder, resulting in a total of 50.8M parameters. Note that while the shape function estimation scales quadratically in the number of features and datapoints, the inference only scales linearly in both.

### 3.3 Training Procedure

We train with SGD on synthetic data priors, a method introduced in Prior-Data Fitted Networks (PFNs) (Müller et al., 2022; Hollmann et al., 2023). These priors are designed to be diverse, facilitating the generation of realistic tabular datasets and enabling extrapolation to real-world data. We use two types of priors for training: (1) Structural Causal Models, which involve sampling random causal graphs and generating data from them, and (2) Gaussian Processes, where random Gaussian Processes are sampled and used to generate data. For more details on the synthetic data generation process, we refer to Appendix D. During training, the synthetic data is randomly split into train and test datasets. To obtain the parameters $\theta$ we minimize a cross-entropy loss between the estimated GAM prediction and ground truth labels on the test dataset $D_{\text{test}}$:

$$\theta^* \in \operatorname{argmin}_\theta \mathbb{E}_{(D_{\text{train}} \cup (x_{\text{test}}, y_{\text{test}})) \sim p(D)} \left[ \mathcal{L}(\tilde{y}_{\text{test}}, y_{\text{test}}) \right] \tag{2}$$

Additional details on the training are given in Appendix E. GAMformer's core contribution is the substitution of the data fitting process of traditional GAM variants with a single forward pass of a pre-trained transformer model, which is presented with data through in-context examples. Consequently, GAMformer replaces the manually crafted fitting procedures used in methods like EBMs (Caruana et al., 2015), where the boosting procedure is restricted to one feature at a time in a round-robin manner, or the joint optimization of all shape functions in NAMs (Agarwal et al., 2021) using SGD. Note that in both traditional GAM fitting and GAMformer, the output of the processes remains the same; a main effects GAM fitted to a given dataset represented by its shape functions. We provide a carbon footprint analysis compared to EBMs in appendix E.1, demonstrating that GAMformer's less costly (in terms of carbon emmission) inference than EBMs offset its training cost after enough model applications.

### 3.4 Higher-order effects

We now describe how GAMformer can be extended to handle higher-orders effects. We extend GAMformer to model higher-order effects, specifically pairwise interactions, by incorporating feature products, resulting in up to $\mathcal{O}(p^2)$ potential features. GAMformer can accommodate this by performing ICL on concatenated original data and higher-order effects, represented as feature vectors in $\mathbb{R}^{p+P}$, where $P$ denotes the number of pair interactions. However, increasing feature dimensions beyond the 100 used in pretraining is problematic and adds complexity to shape function estimation. To mitigate this, we rank the most informative pairs via the FAST method (Lou et al., 2013) and the optimal number of pairs is determined as a hyperparameter through cross-validation during inference.

## 4 Experiments

After pretraining GAMformer on the synthetic datasets, we evaluate it on both illustrative and real-world tasks in 4.1 and 4.4, respectively. Moreover, in 4.2 and 4.3, we highlight its potential in real world applications (assisting in decision-making in a clinical setting by predicting the mortality rate of patients in the intensive care unit (ICU), and estimating house prices). We chose Explainable Boosting Machines (EBMs) (Lou et al., 2012; 2013; Caruana et al., 2015) as our primary GAM baseline because their shape-function modeling approach is most analogous to GAMformer's. Both methods learn fully non-parametric shape functions that can capture sharp discontinuities. In addition, we compare to spline-based GAMs from the `mgcv` package (Wood, 2001) and other strong tabular classification models such as XGBoost (Chen & Guestrin, 2016) and TabPFN (Hollmann et al., 2023) in terms of predictive performance. On the downstream datasets,

---

[1]This embedding is equivariant with respect to input features but not invariant to class ordering due to distinct class encodings in the input layer.

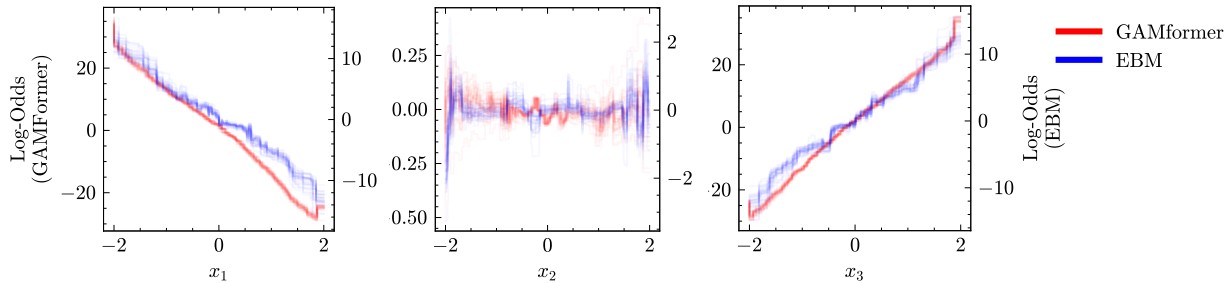

Figure 2: Shape functions derived from GAMformer and EBMs applied to the linear, binary classification problem $f(x_1, x_2, x_3) = \mathbb{I}((-1)x_1 + 0x_2 + x_3 > 0)$. We use a twin y axis with GAMformer and EBM on left and right, respectively. All models shown result from a 30-fold cross-validation over 1500 data points.

differently from EBM and the other baselines, GAMformer requires *only a single forward pass* of the transformer model to estimate the shape functions and construct predictions on the entire test set, without any parameter updates.

## 4.1 Illustrative Examples

Before demonstrating GAMformer on real-world tabular data, we first investigate its behavior on synthetic data where the data-generation process is known. This allows us to validate the effectiveness of GAMformer in capturing the underlying relationships between features and the target variable. All considered examples are binary classification and hence we only show one shape function per class and per feature. In the context of GAMs with a logit link function (used for binary classification), log-odds is the unit of the predictors. Therefore, the shape functions' output values are on the log-odds scale, which are then transformed to overall prediction probabilities after summing via the logistic function. For all metrics reported in the paper, we use ROC-AUC (Receiver Operating Characteristic - Area Under the Curve), a metric which is robust towards imbalanced classes.

**Linear, binary classification.** We begin by comparing GAMformer and EBMs on data generated by the linear, binary classification problem $f(x_1, x_2, x_3) = \mathbb{I}((-1)x_1 + 0x_2 + x_3 > 0)$, where $\mathbb{I}$ is the indicator function. We sample 2000 data points uniformly and independently from the interval [-2, 2] and split the data into 1500 training points and 500 test points. The results, shown in Figure 2, demonstrate that both GAMformer and EBMs accurately estimate the slopes for each feature and achieve a ROC AUC of 1.0 on the test dataset. However, the shape functions learned by GAMformer are noticeably smoother, suggesting that it may have captured some bias towards smoother models during pretraining. Additionally, we compared the effect of varying the number of datapoints

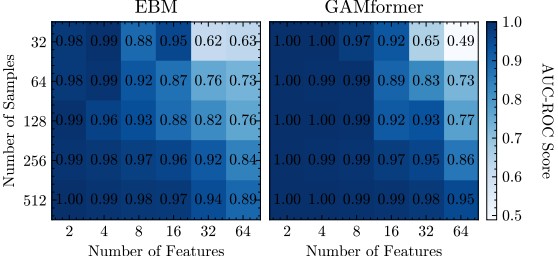

Figure 3: Robustness analysis (linear, binary classification): GAMformer consistently outperforms or matches EBM across various sample sizes and feature counts, showcasing its efficiency.

or features in this example on EBMs and GAMformer in Figure 3. Our findings indicate that GAMformer consistently outperforms EBMs across various sample sizes and number of features.

**Polynomial, binary classification.** To further validate the robustness of GAMformer, we evaluate it on data generated by a more complex function $f(x_1, x_2) = \mathbb{I}(x_1 + x_2^2 > 0)$. The experimental setup remains the same as for the logistic regression case. The results in Figure 4 show that both GAMformer and EBMs successfully capture the quadratic relationship in $x_2$ and the linear contribution of $x_1$ up to $x_1 \leq 0$. For $x_1 > 0$, $f$ always predicts true, resulting in a constant contribution. Consistent with the previous experiment, GAMformer produces smoother shape functions. Again both models achieve an ROC AUC of 1.0 on the test dataset.

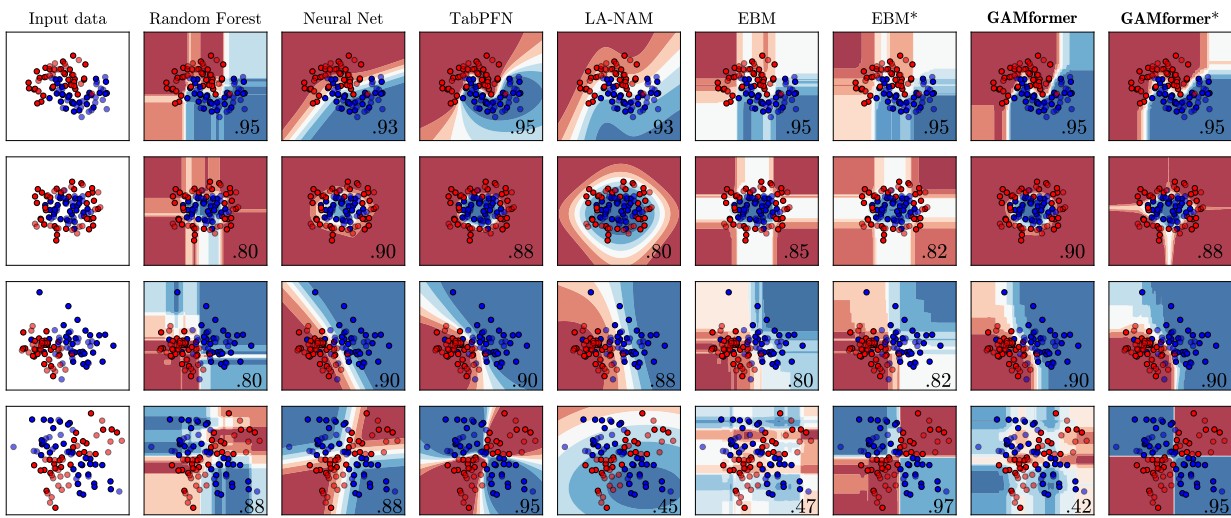

Figure 5: Visualization of classification boundaries for various baseline classifiers and GAMformer on scikit-learn dataset examples (Pedregosa et al., 2011), in the lower right corner we show the ROC-AUC on a validation split. Due to the absence of higher-order feature interaction terms in both GAMformer and EBM (main effects), the 'XOR' dataset (bottom row) is not accurately modeled by them. Incorporating second-order effects solves the problem (EBM* and GAMformer*).

**Classification Boundaries.** We visualize the classification boundaries of GAMformer compared to TabPFN and EBM on the scikit-learn (Pedregosa et al., 2011) test datasets in Figure 5. We find that GAMformer performs similarly to TabPFN and EBMs on most of the example datasets. LA-NAM (Bouchiat et al., 2024) (main effects only), a Bayesian version of NAMs (Agarwal et al., 2021), provides good uncertainty estimates despite exhibiting slightly worse predictive performance. It is worth noting that GAMformer, EBM and LA-NAM struggle with accurately modeling

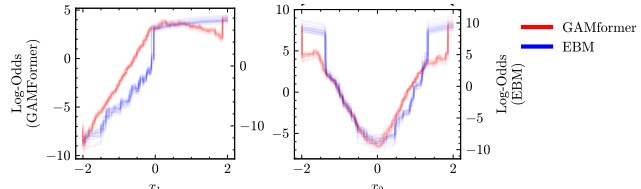

Figure 4: (a) Shape functions derived from GAMformer and EBMs applied to the polynomial, binary classification problem $f(x_1, x_2) = \mathbb{I}(x_1 + x_2^2 > 0)$. All models result from a 30-fold cross-validation over 1500 data points are shown.

the 'XOR' dataset (bottom row) due to the absence of higher-order feature interaction terms in these models. This is resolved by incorporating second-order effects (EBM* and GAMformer*; see Section 3.4 for details), allowing them to effectively learn the non-linear decision boundary of the 'XOR' function.

## 4.2 Case Study (Classification): Intensive Care Unit Mortality Risk

In this case study, we examine shape functions derived from GAMformer and EBMs (main effects only) using the MIMIC-II dataset (Lee et al., 2011), a publicly available critical care dataset for predicting mortality risk based on various demographic and biophysical indicators. Our analysis focuses on four key clinical variables: Age, Heart Rate (HR), PFratio (PaO2/FiO2 ratio), and Glasgow Coma Scale (GCS), as shown in Figure 6 (remaining variables in Figure 11). See Figure 12 and Figure 13 in Appendix G for the shape functions of the remaining features on the MIMIC-III dataset.

For **Age**, the GAMformer shape function shows a steady increase in the log-odds of adverse outcomes with advancing age, stabilizing at older ages. The data density plot reveals a higher concentration of data points in middle age, with fewer at the extremes. The shape function exhibits less variance where data is denser, indicating the model's reliability in these regions. Overall, the shape function highlights increased risk in elderly patients due to declining physiological reserves and multiple chronic conditions.

**Heart Rate (HR)** exhibits a complex relationship with adverse outcomes. Both GAMformer and EBMs capture a U-shaped risk profile, indicating increased risk at very high and very low heart rates, underscoring the importance of maintaining HR within a normal range.

**PFratio**, a lung function and oxygenation efficiency measure, shows a steep risk increase as values decrease. Lower PFratio values, critical in diagnosing and managing conditions like Acute Respiratory Distress Syndrome (ARDS), indicate worse lung function. Notably, both models display a sharp drop in risk at a PFratio of approximately 325, likely an artifact from data preprocessing where missing values were imputed at the mean, previously pointed out by Chen et al. (2023) for MIMIC-2. In healthcare, missing values often suggest healthier patients, as data collection was deemed unnecessary by professionals. Here, patients with missing PFratio values, representing the majority, have lower risk than those with collected values. GAMformer precisely isolates these missing value patients, demonstrating its potential to detect data processing artifacts better than prior GAM algorithms.

For the **Glasgow Coma Scale (GCS)**, which measures the level of consciousness, there is a strong negative correlation with adverse outcomes. Lower GCS scores, indicating reduced consciousness, are associated with significantly higher mortality risk. Our findings show that GAMformer effectively handles categorical data, identifying patterns similar to those detected by EBMs.

### 4.3 Case Study (Regression): California Housing

In addition to the multi-class classification GAMformer, we trained another GAMformer model for regression. Crucially, we did not have to modify the synthetic data used during training: it remained identical to that used in classification, with the simple mofification that we remove the label assignment usually included at the end of data generation pipeline. Using an identity link function and MSE loss, we trained a new model and evaluated it on the California Housing dataset (Kelley Pace & Barry, 1997). The California Housing Dataset, derived from 1990 California Census data, contains 20,640 samples with housing attributes like median income, house age, room counts, and geographic coordinates for predicting median house values across census block groups. As shown in Figure 7, GAMformer and EBMs consistently capture similar trends across features. For **Housing Median Age** and **Median Income**, both models exhibit a clear positive correlation with the house price, with GAMformer providing smoother general trends while EBM captures more detailed variations. The **Latitude** and **Longitude** plots reveal complex, non-linear relationships where GAMformer accurately captures the housing price spikes for San Francisco (Lat. 37, Long. -122) and Los Angeles (Lat. 33, Long. -118). The other shape functions are shown in Figure 14.

### 4.4 Multi-class Classification on OpenML Tabular Datasets

To assess the transferability of pretraining on synthetic data to real-world tabular data, we evaluate GAMformer's performance on the test datasets from TabPFN (Hollmann et al., 2023), which include up to 2000 datapoints (see Appendix B for dataset details). Figure 8 reports Critical Difference (CD) diagrams (Demšar, 2006) showing the average rank across datasets for each method, with statistically tied methods grouped by horizontal bars. Our method outperforms EBM when using only main effects. With pair effects, both GAM-

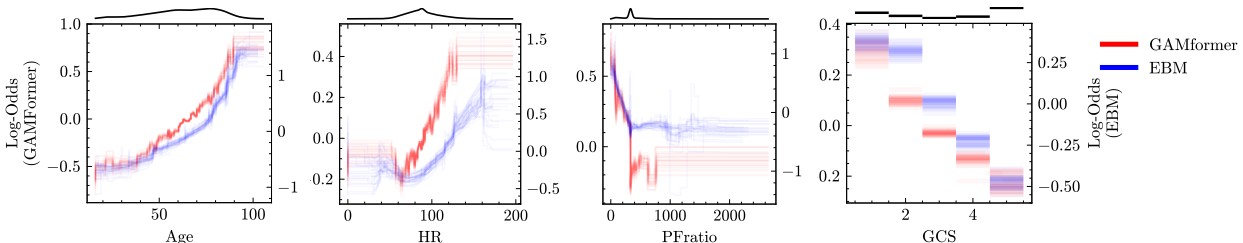

Figure 6: *Classification*: Shape functions derived from GAMformer and EBMs applied to the MIMIC-II dataset for critical clinical variables. The data density plot is shown above each figure. The results are based on 30 models for both GAMformer and EBMs, each fitted on 10,000 randomly selected data points.

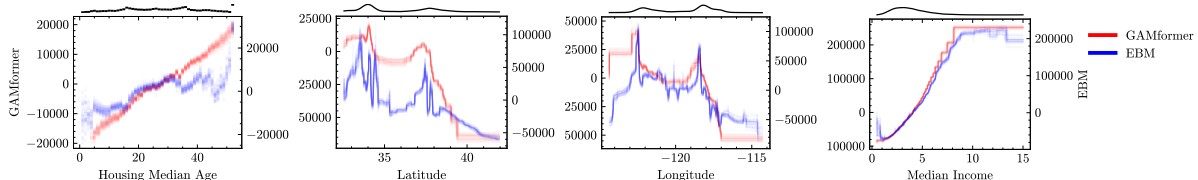

Figure 7: *Regression:* GAMformer vs. EBM shape functions for key variables in the California Housing dataset. Each subplot displays data density (top) and learned shape functions (bottom). Results are averaged over 30 models, each evaluated (GAMformer)/trained (EBM) on 10,000 randomly sampled data points. While EBM achieved slightly lower error (RMSE: 58004.48 ± 39.99 vs. 68220.5 ± 125.96), GAMformer demonstrates comparable interpretability and captures similar trends, validating its effectiveness in producing meaningful, interpretable models.

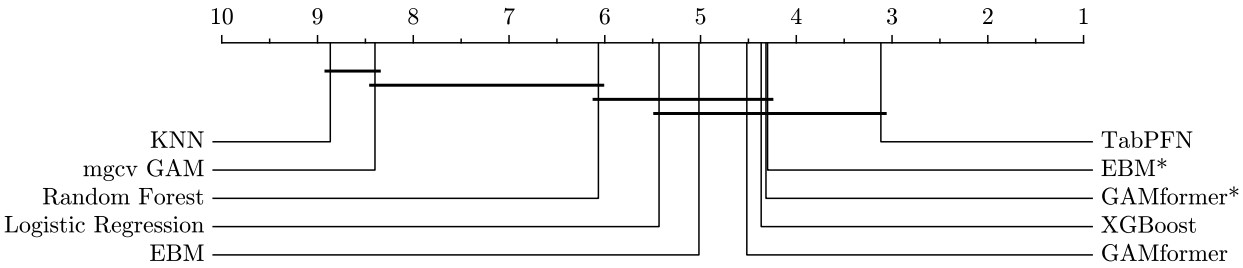

Figure 8: Critical Difference diagram demonstrating GAMformer's competitive performance against state-of-the-art baselines across various tabular datasets. Lower ranks indicate superior performance; connected algorithms are not statistically significantly different ($p = 0.05$).

former* and EBM* show slight improvements, matching XGBoost's performance. We also compare against spline-based GAMs from the `mgcv` R library (Wood, 2001), demonstrating significantly better performance for GAMformer.

The small performance gap between XGBoost and GAMformer indicates that main effects-only GAMs sacrifice less predictive power than commonly assumed, making their interpretability advantages particularly valuable for many applications.

## 5  Limitations & Broader Impact

**Limitations.** While GAMformer introduces a novel approach to estimating Generalized Additive Models (GAMs), it is important to acknowledge its current limitations. This work primarily focuses on main and second-order effect GAMs and does not account for higher-order interactions, which are addressed in other GAM implementations, such as EBMs (Lou et al., 2013; Nori et al., 2019; Chang et al., 2021). Future research could explore incorporating these interactions to enhance the model's expressiveness and predictive capabilities. Another limitation of the current GAMformer model is its difficulty in improving predictions when presented with datasets that exceed twice the size of the data it saw during training (c.f. Figure 9). This issue is related to the well-known challenge of length extrapolation in sequence-to-sequence models, including transformers (Grazzi et al., 2024; Zhou et al., 2024). This is a known challenge for the current generation of tabular foundation models, and promising solutions are emerging. Future work could incorporate more scalable architectures, such as those using linear attention (e.g., TabFlex (Zeng et al., 2025)) or modified attention patterns (e.g., TabICL (Qu et al., 2025)), to mitigate these issues.

**Broader Impact.** As a versatile machine learning model for tabular data, GAMformer offers both positive and negative societal impacts. Positively, it can generate novel insights in fields like medicine, enhancing disease diagnosis and treatment. However, it can also be misused to not mitigate but exploit biases, such as adjusting insurance premiums based on ethnicity, leading to discrimination.

# 6 Conclusion

In this paper, we introduce GAMformer, the first tabular foundation model for Generalized Additive Models (GAMs) that bridges the interpretability gap in foundation model approaches to tabular data. By leveraging in-context learning to estimate shape functions in a single forward pass, GAMformer addresses the fundamental incompatibility between traditional iterative GAM methods and the foundation model paradigm. Our approach uses non-parametric, binned representations of shape functions, enabling interpretable modeling while maintaining the efficiency advantages of foundation models. Extensive experiments demonstrate that GAMformer achieves comparable accuracy to leading GAM variants across various classification benchmarks while exhibiting robustness to label noise and class imbalance, despite being trained exclusively on synthetic data.

GAMformer offers a novel approach that moves from iterative optimization methods to foundation model techniques for interpretable tabular modeling. Our case studies on the MIMIC-II dataset and California Housing dataset demonstrate that GAMformer's shape functions provide qualitative insights and can uncover dataset flaws similar to state-of-the-art GAM methods, confirming that interpretability is preserved in the transition to foundation models across both classification and regression tasks. This work opens a new research direction at the intersection of interpretable machine learning and foundation models, with immediate applications in safety-critical domains where both high performance and transparency are required. Future research can build on this foundation model paradigm for interpretable ML, exploring scalable architectures and extending the approach to other interpretable model families beyond GAMs. Finally, one can view GAMs as predicting flexible shape functions for a structural causal model with only direct effects of features on the outcome. Future work could explore how the approach of GAMformer can be combined with recent progress in causal foundation models (Robertson et al., 2025) to estimate the full structural causal model.

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

# A   Generalized Additive Models: Extended Related Work

As with many families of machine learning algorithms, the differences among GAM algorithms lie in (a) the functional form of the shape functions $f_i$, (b) the learning algorithm used for their estimation and (c) regularity assumptions and regularization. Two important properties that all GAMs share are (1) the ability to learn non-linear transformations for each feature and (2) additively combining these shape functions (prior to applying the link function) to create modularity that aids interpretability by allowing users to examine shape functions one-at-a-time.

Typically, GAMs have relied on splines and backfitting algorithms for estimation (Hastie & Tibshirani, 1987), with subsequent works focusing on improving efficiency and stability through penalized regression splines (Wood, 2003) and fast, stable fitting algorithms (Wood, 2001). Spline-based GAMs are typically fitted using the backfitting algorithm, an iterative procedure that starts with initial estimates of the smooth functions for each predictor variable. The algorithm then repeatedly updates each function by fitting a weighted additive model to the residuals of the other functions until convergence is achieved. The weights are determined by the current estimates of the other functions and the link function in the case of generalized additive models.

Modern approaches leverage machine learning advances. Explainable Boosting Machines (EBMs) (Lou et al., 2012; 2013; Caruana et al., 2015) model the shape functions using decision trees, which are fitted using a variant of gradient boosting called cyclic gradient boosting. The model iteratively learns the contribution of each feature and interaction term in a round-robin fashion, using a low learning rate to ensure that the order of features does not affect the final model. This cyclic training procedure helps mitigate the effects of colinearity among predictors by providing opportunity for data-driven credit attribution among the features while preventing multiple counting of evidence. EBMs are also popular because they can accurately capture steps in the shape functions, which is important for modeling discontinuities in data, such as treatment effects in medical data.

More recently, Neural Additive Models (NAMs) (Agarwal et al., 2021) and follow up works (Chang et al., 2021; Dubey et al., 2022; Radenovic et al., 2022; Xu et al., 2022; Enouen & Liu, 2022; Bouchiat et al., 2024) use multilayer perceptrons (MLPs), as non-linear transformations, to model the shape functions $f_i$. As a result, NAMs can be optimized using variants of gradient descent by leveraging automatic differentiation frameworks.

Finally, GAMs have also found applications in time-series forecasting, with models such as Prophet (Taylor & Letham, 2018) and NeuralProphet (Triebe et al., 2021). Interestingly, the 1-layer versions of the recently proposed Kolmogorov-Arnold Networks (KANs) (Liu et al., 2024) may be viewed as GAMs with spline based shape functions.

# B   Dataset Details

In this section, we provide details on the datasets used in our empirical evaluations of GAMformer.

As test dataset, we used the 30 datasets used in Hollmann et al. (2023) which were obtained from OpenML (Vanschoren et al., 2014). These were chosen because they contain up to 2000 samples, 100 features and 10 classes; they are detailed in Table 1.

# C   Properties of GAMformer

## C.1   Data Scaling

To assess GAMformer's ability to generalize to datasets containing more datapoints than it saw during training, i.e. larger context sizes, we conducted an experiment that varied the number of training data points and evaluated the impact on ROC-AUC performance using a consistent validation split. To ensure the robustness of our findings, we sampled training datasets three times with replacement for each training size. The results in Figure 9 demonstrate that GAMformer's ROC-AUC improves across datasets when the number of training examples is up to twice the number of training examples seen during training. For

Table 1: Test dataset names and properties, taken from Hollmann et al. (2023). Here *did* is the OpenML Dataset ID, *d* the number of features, *n* the number of instances, and *k* the number of classes in each dataset.

| did | name | d | n | k | did | name | d | n | k |
|-----|------|---|---|---|-----|------|---|---|---|
| 11 | balance-scale | 5 | 625 | 3 | 1049 | pc4 | 38 | 1458 | 2 |
| 14 | mfeat-fourier | 77 | 2000 | 10 | 1050 | pc3 | 38 | 1563 | 2 |
| 15 | breast-w | 10 | 699 | 2 | 1063 | kc2 | 22 | 522 | 2 |
| 16 | mfeat-karhunen | 65 | 2000 | 10 | 1068 | pc1 | 22 | 1109 | 2 |
| 18 | mfeat-morphological | 7 | 2000 | 10 | 1462 | banknote-authentication | 5 | 1372 | 2 |
| 22 | mfeat-zernike | 48 | 2000 | 10 | 1464 | blood-transfusion-... | 5 | 748 | 2 |
| 23 | cmc | 10 | 1473 | 3 | 1480 | ilpd | 11 | 583 | 2 |
| 29 | credit-approval | 16 | 690 | 2 | 1494 | qsar-biodeg | 42 | 1055 | 2 |
| 31 | credit-g | 21 | 1000 | 2 | 1510 | wdbc | 31 | 569 | 2 |
| 37 | diabetes | 9 | 768 | 2 | 6332 | cylinder-bands | 40 | 540 | 2 |
| 50 | tic-tac-toe | 10 | 958 | 2 | 23381 | dresses-sales | 13 | 500 | 2 |
| 54 | vehicle | 19 | 846 | 4 | 40966 | MiceProtein | 82 | 1080 | 8 |
| 188 | eucalyptus | 20 | 736 | 5 | 40975 | car | 7 | 1728 | 4 |
| 458 | analcatdata_authorship | 71 | 841 | 4 | 40982 | steel-plates-fault | 28 | 1941 | 7 |
| 469 | analcatdata_dmft | 5 | 797 | 6 | 40994 | climate-model-... | 21 | 540 | 2 |

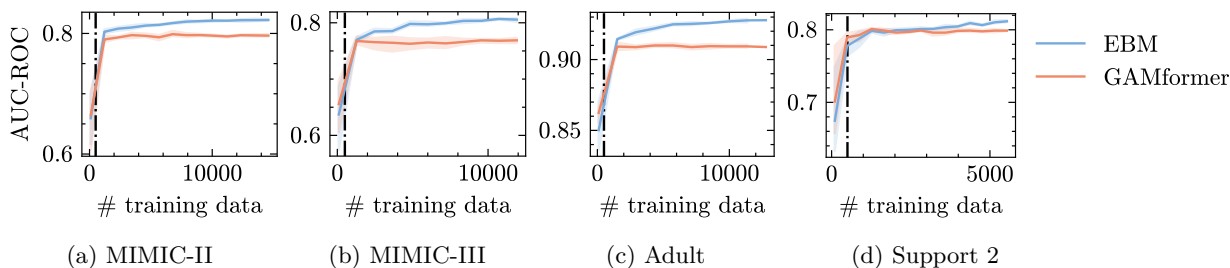

(a) MIMIC-II  (b) MIMIC-III  (c) Adult  (d) Support 2

Figure 9: Demonstration of the ability of GAMformer to scale beyond the datapoints seen during training while leveraging the additional data points to increase its performance. The dashed vertical line denotes the number of in-context examples seen during training (500).

comparison, we also evaluated the performance of EBMs under the same conditions. While EBMs also exhibited improvements in ROC-AUC with increased training data, they achieved higher accuracy when provided with a larger number of examples. This observation highlights a limitation of GAMformer in its ability to fully leverage additional training samples.

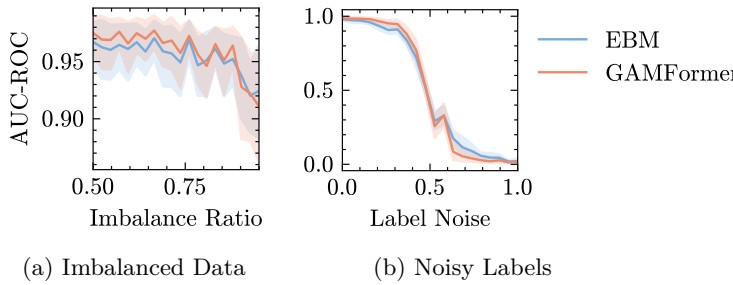

(a) Imbalanced Data  (b) Noisy Labels

Figure 10: Comparison of GAMformer and EBMs in terms of (a) performance on class imbalanced data and (b) robustness to noisy labels. The shaded areas represent the 5% and 95% confidence intervals estimated using 1000 bootstrap samples.

## C.2 Class Imbalance

To compare GAMformer's sensitivity to class imbalance with that of EBMs, we conduct the following analysis. First, we sample 300 data points from two centroids in a 20-dimensional feature space, creating a binary classification problem. We then vary the ratio of the two classes to introduce increasing levels of imbalance in the sampled data. Next, we split the data into train and test sets using a 75% to 25% split and evaluate the performance using the AUC-ROC metric. We repeat the experiment 10 times for each data ratio. Our results are shown in Figure 10a, the shaded area are the 5%, 95% confidence intervals estimated using 1000 bootstrap samples. We see that GAMformer performs on average better than EBMs in this setting and shows no inherent sensitivity to class imbalance.

## C.3 Noise Robustness

To gain a deeper understanding of GAMformers' sensitivity to noisy or incorrect labels, we conducted an experiment similar to the one described in Appendix C.2. We generated 300 data points and randomly perturbed the labels in the train split with increasing probability (75%, 25% train/test split), repeating each experiment 10 times. Figure 10b illustrates our findings. Once again, we observed that GAMformer exhibits a sensitivity to noisy labels comparable to that of EBMs.

# D   Synthetic Data Priors

We use the same synthetic data generation process proposed in Prior-Data-Fitted Networks (PFNs) (Hollmann et al., 2023; Müller et al., 2022) and provide a brief summary of the process.

TabPFN is trained on two synthetic data priors, which are mixed during training.TabPFN introduced a synthetic data prior based on Structural Causal Models (SCMs). SCMs are particularly suitable for modeling tabular data as they capture causal relationships between columns, a strong prior in human reasoning. An SCM comprises a set of structural assignments (mechanisms) where each mechanism is defined by a deterministic function and a noise variable, structured within a Directed Acyclic Graph (DAG). The causal relationships are represented by directed edges from causes to effects, facilitating the modeling of complex dependencies within the data. To instantiate a PFN prior based on SCMs, one defines a sampling procedure to create supervised learning tasks. Each dataset is generated from a randomly sampled SCM, including its DAG structure and deterministic functions. Nodes in the causal graph are selected to represent features and targets, and samples are generated by propagating noise variables through the graph. This process results in features and targets that are conditionally dependent through the DAG structure, capturing both forward and backward causation (Hollmann et al., 2023). This allows for the generation of diverse datasets.

The second prior samples of synthetic data using Gaussian Processes (GPs) (Rasmussen & Williams, 2006) with a constant mean function and a radial basis function (RBF) kernel to define the covariance structure. Hyperparameters such as noise level, output scale, and length scale are sampled from predefined distributions to introduce variability. Depending on the configuration, input data points can be sampled uniformly, normally, or as equidistant points and the target column is generated by passing the input data through the GP. This prior gives the model the ability to learn smoother functions.

For multi-class prediction, scalar labels are transformed into discrete class labels by partitioning the scalar values into intervals corresponding to different classes, ensuring the synthetic data is suitable for imbalanced multi-class classification tasks.

Finally, both priors are combined by sampling batches of data from each prior with different probabilities during training. In all of our experiments we sampled from the SCM and GP prior with probability 0.96 and 0.04, respectively.

# E   Training Details

In GAMformer, we used a transformer model with 12 hidden layers, 512 embedding size and 4 heads per attention. To bin the shape functions and all features we used 64 bins. For training, we use the

AdamW (Loshchilov & Hutter, 2019) optimizer ($\beta_1 = 0.9$) and cosine learning rate schedule with initial learning rate of 3e-5, 20 warm up epochs and minimum learning rate of 1e-8 for 25 days on a A100 GPU with 80Gb of memory. We used mixed precision training. Each epoch (arbitrarily) consists of 65536 synthetic datasets; the model trained for 1800 epochs, meaning it saw over 100M synthetic datasets. We used a batch size of 8, that we doubled at epoch 20, 50, 200 and 1000. Each synthetic dataset consisted of 500 samples that were split into training and test portions using using a uniform sampling of the training fraction, and used a number of features drawn uniformly between 1 and 10.

### E.1 Carbon Cost Analysis

GAMformer pre-training carbon cost is offset after **7.7 million forward passes** compared to retraining an EBM.

**Setup:** GAMformer (A100 GPU, 300W), EBM (AMD EPYC 9334, 210W). Carbon intensity: 0.350 kg $CO_2$/kWh (EPA eGRID 2023).

**Pre-training:** 600 hours $\times$ 0.3 kW = 180 kWh = 63 kg $CO_2$

**Per-run costs:**

- EBM: 0.645s runtime = $1.32 \times 10^{-5}$ kg $CO_2$

- GAMformer: 0.170s runtime = $4.96 \times 10^{-6}$ kg $CO_2$

**Break-even:** Solving $63 = N \times (1.32 \times 10^{-5} - 4.96 \times 10^{-6})$ yields $N = 7.7$ million runs.

We note that 7.7 million runs are not uncommon for foundation models; e.g., TabPFN has been downloaded about 2 million times (see the download statistics), and the standard TabPFN demo Colab that most users likely execute already runs dozens of models.

## F  Higher-order effects

To handle higher-order effects, we compute the best pairs with the FAST algorithm (Lou et al., 2013) and evaluate GAMformer on the top pairs using the following ratios of features:

$$\mathcal{P} = [0.01p, 0.05p, 0.1p, 0.2p, 0.4p, 0.8p, 0.9p]$$

where we recall that $p$ denotes the number of features. We round off each ratio to determine the number of target pair features, evaluate performance on hold-out validation data from the training set, and select the number of pairs with the best validation performance. The model is then fitted on the entire training dataset. This involves doing $|\mathcal{P}| + 1$ forward passes, which is unproblematic as doing one forward pass is very fast, even on a CPU. One could also vectorialize all computations which we do not do given the low fitting time.

## G  Shape Functions

In this section, we show complementary results on the shape functions estimates from GAMformer and EBM (main effects only) on the MIMIC-II (Lee et al., 2011) (complementary to the plots in Figure 6) and on the MIMIC-III datasets.

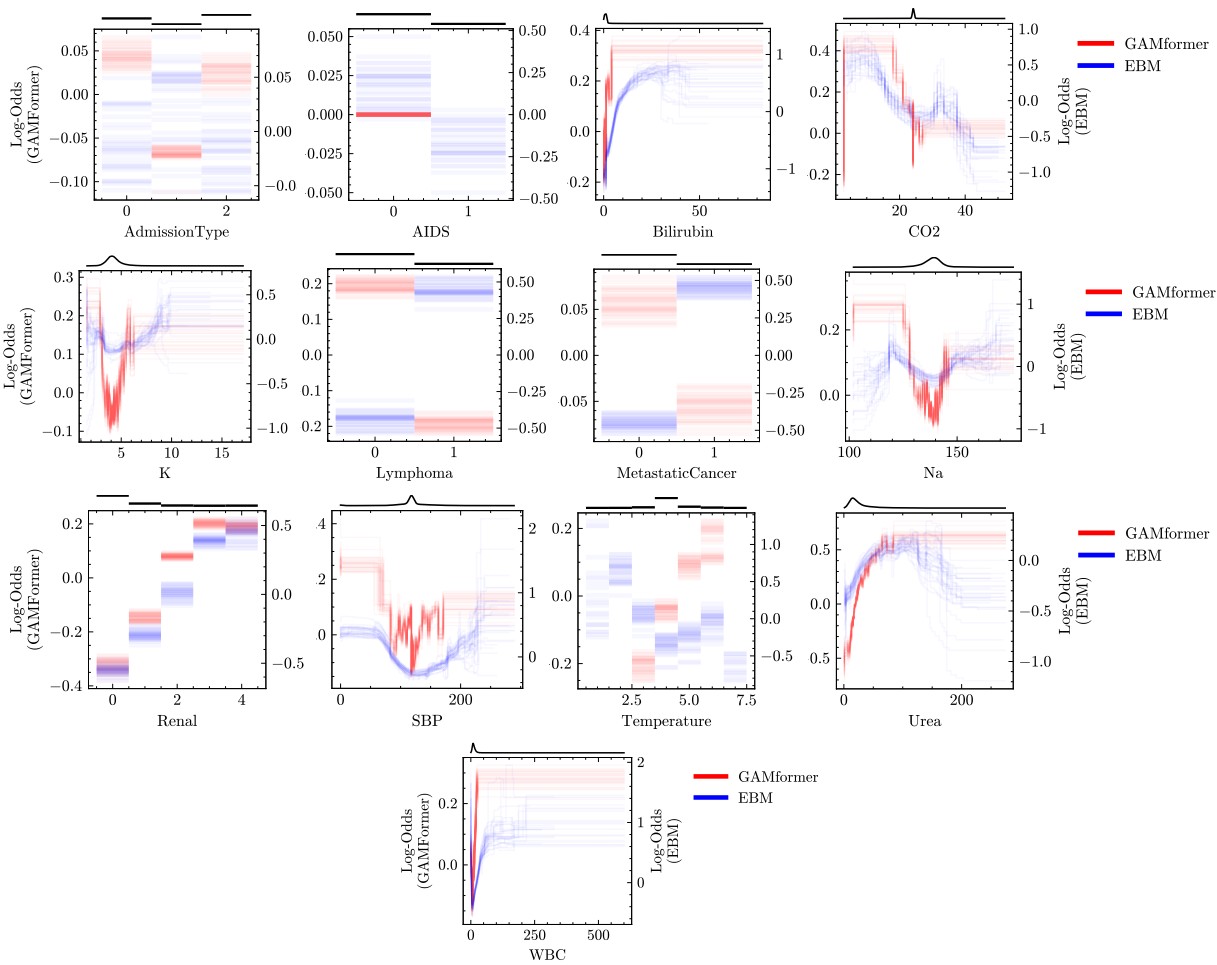

Figure 11: The remaining shape functions derived from GAMformer and EBMs on the **MIMIC-II dataset** for critical clinical variables. The plot above each figure shows the data density. There are interesting differences between the EBM and GAMformer shape plots for several of the categorical variables.

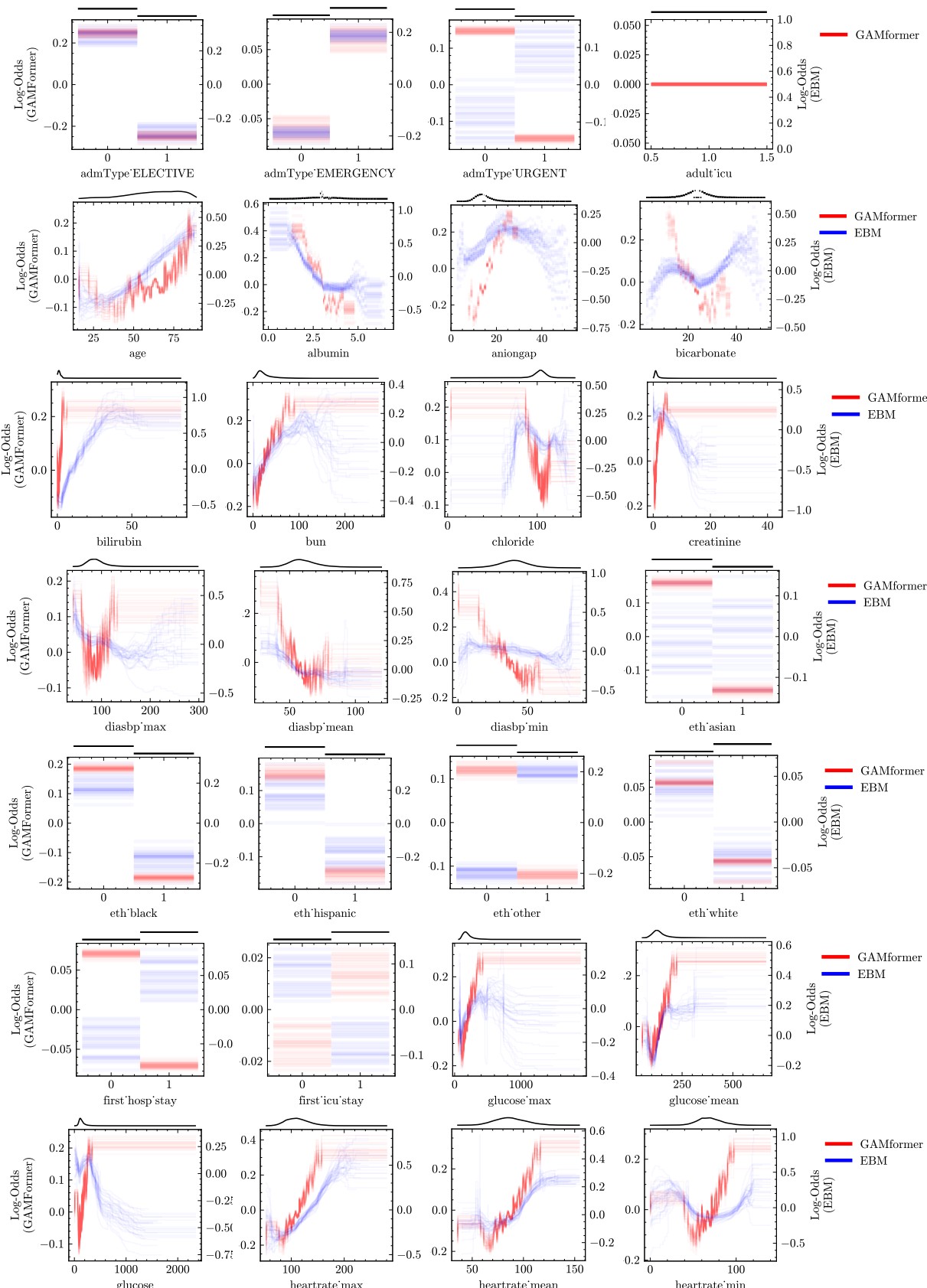

Figure 12: The shape functions derived from GAMformer and EBMs on the **MIMIC-III dataset** for critical clinical variables. The plot above each figure shows the data density. The results are based on 30 models for both GAMformer and EBMs, each fitted on 10,000 randomly selected data points. There are interesting differences between the EBM and GAMformer shape plots for several of the categorical variables. Although different GAM algorithms do not usually learn identical functions, we are investigating to better understand these differences.

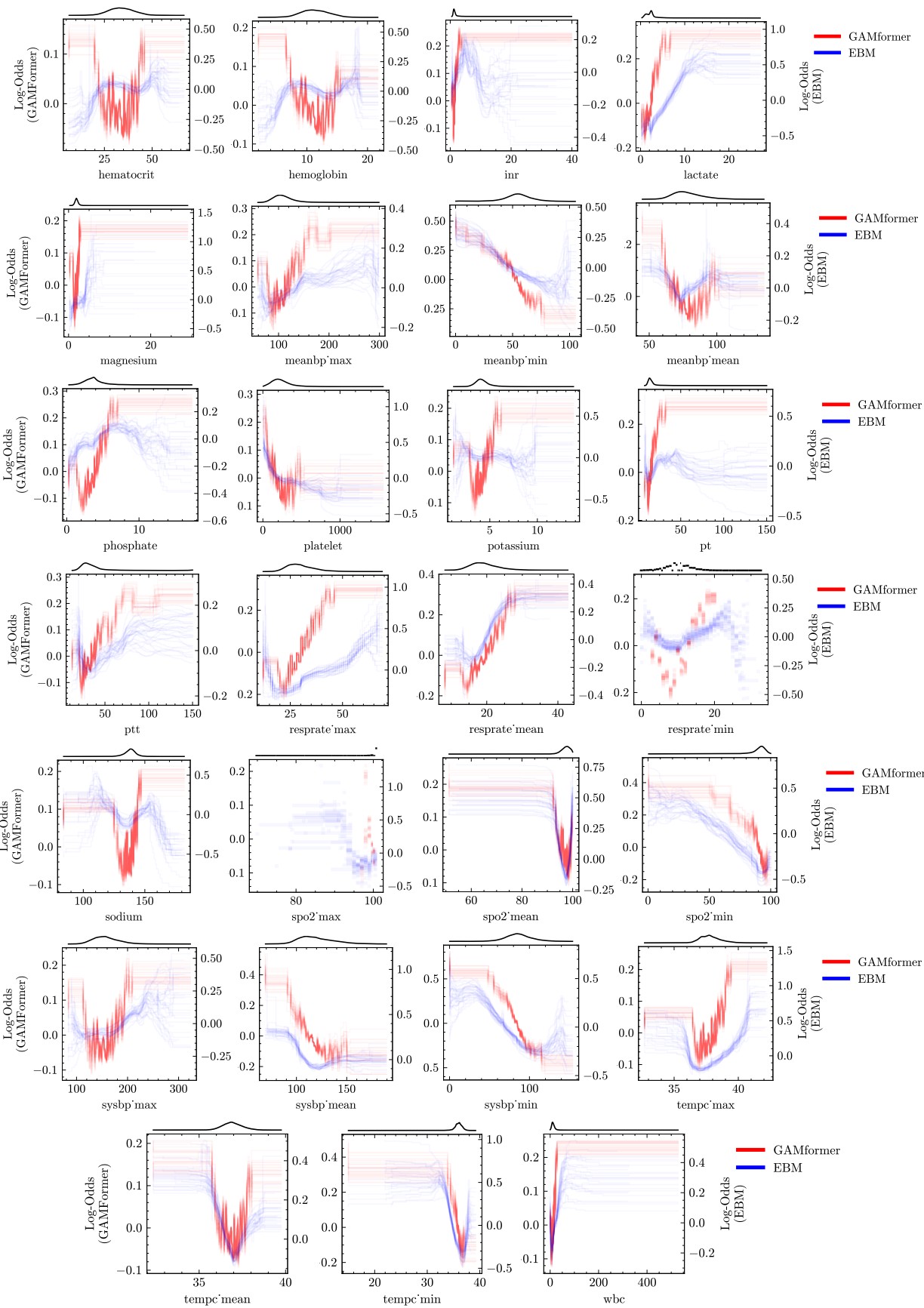

Figure 13: The remaining shape functions derived from GAMformer and EBMs applied to the **MIMIC-III dataset** for critical clinical variables. The plot above each figure shows the data density in the training set. The results are based on 30 models for both GAMformer and EBMs, each fitted on 10,000 randomly selected data points.

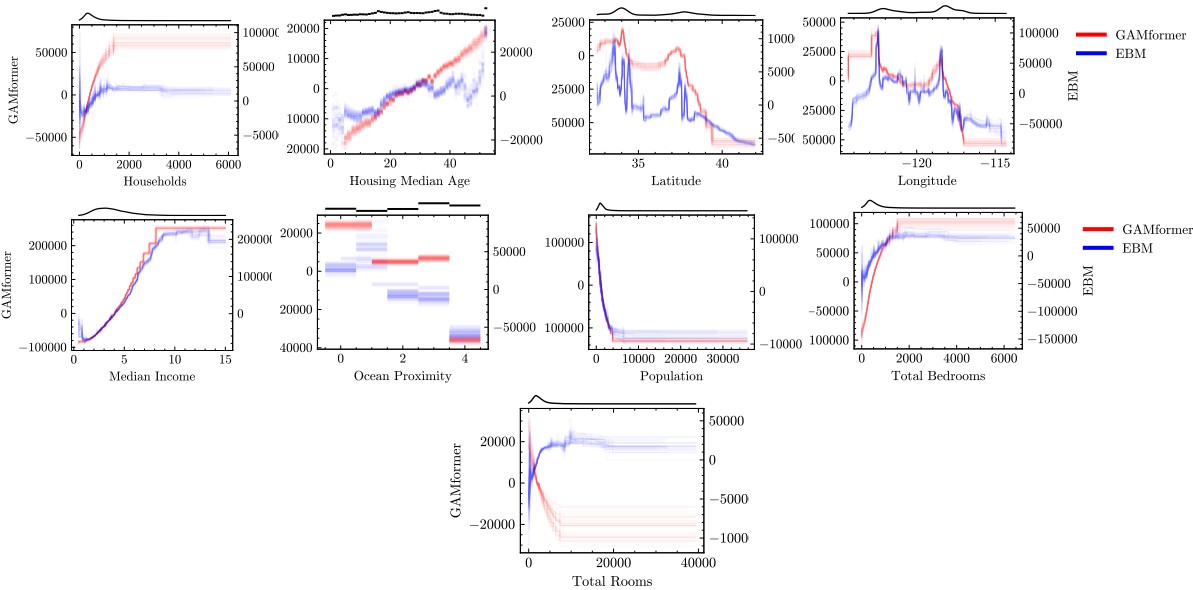

Figure 14: The remaining shape functions derived from GAMformer and EBMs on the **California Housing** for critical clinical variables. The plot above each figure shows the data density.

