# OpenReview forum: "GAMformer: Bridging Tabular Foundation Models and Interpretable Machine Learning"
_TMLR — Rejected by TMLR_

### Review · Reviewer_quL9 · 2026-02-25

**Summary Of Contributions:**

Before evaluating a GAM on test data, typical approaches require training the GAM on training data from the same distribution as the test data, so as to approximate the shape functions. Instead, the proposed method uses a transformer, pretrained on some syntactic data, to approximate the shape functions in its forward loop, without requiring fine-tuning on the task-specific training data. Experimental results show that the resulting model, called GAMformer, behaves as well as those trained on task-specific data.

**Audience:**

Yes

**Audience Explanation:**

The idea of using a transformer pre-trained on synthetic data to estimate shape functions without task-specific fine-tuning may be relevant to researchers working on GAMs.

That said, I am not yet convinced by the motivation as currently presented. My understanding is that training a GAM is often not very expensive in practice (please correct me if this assumption is inaccurate). If that is the case, then the paper should more clearly explain why avoiding GAM training is an important goal, especially given the cost of pre-training a transformer on synthetic data.

**Broader Impact Concerns:**

None.

**Claims And Evidence:**

No

**Claims Explanation:**

This belongs to an empirical paper, so the claims should be supported by sufficient empirical evidence.
However,
1. when illustrating the shape functions, there are only comparisons with the shape functions learned by EBM, rather than other standard baselines such as NAM.
2. when comparing the performance of downstream tasks (e.g., classification) against baselines, there are only qualitative results, i.e., Figure 5, rather than quantitative results such as comparison in classification accuracy.

**Requested Changes:**

1. **Improve the clarity and quality of the writing throughout the paper:** The current writing makes the paper difficult to follow in several places. For example, the introduction section does not provide a brief explanation of GAMs or shape functions. As a result, readers may finish the introduction without a clear understanding of what GAMs are, what shape functions are, and why tabular foundation models are a natural tool for this problem.

2. **Improve the method clarification:** In Sec. 3.1, $n_{bin}$ is used before the term "bin" is clearly defined, and the prediction procedure is not explained clearly enough.

3. **Add empirical evidence:** Include the evidence mentioned in my response to "Are the claims made in the submission supported by accurate, convincing and clear evidence?"

4. **Clarify and better justify the motivation:** If training a task-specific GAM is relatively cheap compared with pre-training a transformer, then the paper should explain more clearly why this tradeoff is worthwhile in practice, and in which settings GAMformer offers a meaningful advantage.

---

### Review · Reviewer_Rdhp · 2026-02-26

**Summary Of Contributions:**

This paper addresses the challenge of interpretable machine learning, specifically focusing on the opacity issues associated with tabular foundation models. To tackle this challenge, it introduces a novel architecture named GAMformer, which is a tabular foundation model pretrained on synthetic data. GAMformer is designed to facilitate in-context learning by generating binned univariate shape functions in a single forward pass, thus eliminating the need for iterative backfitting.

**Audience:**

No

**Audience Explanation:**

Given the current state of the paper, which I believe needs revision, I am uncertain whether the question is relevant at this time.

**Broader Impact Concerns:**

No such concerns.

**Claims And Evidence:**

No

**Claims Explanation:**

No, this paper does not meet the criteria of interpretable machine learning (see discussions below).

**Requested Changes:**

**Strengths:**

The limited interpretability of current tabular foundation models poses a significant challenge, especially for sensitive applications that require transparency. The GAMformer architecture provides a solution to this issue. Conceptually, Generalized Additive Models (GAMs) can be considered interpretable under certain conditions, so integrating in-context learning with GAMs is a valid approach.

**Weaknesses:**

The interpretability of GAMs in this context appears to rely on just two properties:
* Additive Transparency: Each shape function $ g_j $ contributes independently.
* Shape Transparency: Each $ g_j $ can be visualized.

This is insufficient. The discussion does not address the sparsity of these models, as there are no constraints on the number of shape functions $ p $. Additionally, the local interpretability of shape functions is not examined, particularly with regard to monotonicity conditions. There is also a lack of robustness guarantees since shape functions can exhibit instability. Finally, the uniqueness of the models is not acknowledged; due to the Roshodomon effect, different GAMs can fit the same data equally well.

While I am not an expert in Transformer architectures, it seems that GAMformer is merely a differentiable two-step pipeline for GAM estimation. While the transformer can indeed learn to output meaningful shape functions, there is no guarantee that these shapes are faithful, stable, and identifiable from an interpretability perspective. Although binning can help preserve interpretability, using 64 bins per feature may result in shapes that are too complex to comprehend. Furthermore, it appears that no regularization is applied to enforce simplicity.

**Request for Changes:**

I believe the central misconception of this paper is that "additivity does NOT equate to interpretability." In general, interpretability requires qualities such as sparsity, smoothness or monotonicity, stability, and ideally, semantic alignment.

Here are the points that this paper should address:

1. **Sparsity Mechanism:** The current model produces a shape function for every feature. If $p = 100$, this results in 100 shape plots, which far exceed the cognitive limit suggested by Miller (approximately $ 7 \pm 2$) and observed in various cognitive psychology experiments. Since sparsity is crucial in interpretable machine learning (see, for example, Lage et al., 2019), a revised version of the GAMformer architecture should incorporate a sparsity mechanism, such as an $L_0$ constraint on the GAM to facilitate learning.

2. **Monotonicity and Smoothness Constraints:** The shapes produced can be arbitrarily jagged, which undermines interpretability, even when $p$ is small. Monotonicity and smoothness are key criteria in interpretable machine learning (see, for instance, Marques-Silva et al., 2021). Ideally, a regularization mechanism that captures desirable criteria such as monotonicity or smoothness should be included in the learning of shape functions.

3. **Stability Analysis:** Different runs of the model may yield varying shapes. If small perturbations can drastically change these shapes, the overall interpretability of the model becomes questionable. Again, a regularization mechanism may be required to provide stable models, which should be confirmed through intensive experiments.

4. **Cognitive Interpretability:** The paper does not address whether humans can actually understand these shapes. A revised version should focus on this question. Additionally, the related work section should include a comprehensive state-of-the-art review in interpretable machine learning, along with the commonly accepted criteria for what constitutes an interpretable model.


**References**

Lage, I., Chen, E., He, J., Narayanan, M., Kim, B., Gershman, S. J., & Doshi-Velez, F. (2019, October). Human evaluation of models built for interpretability. In Proceedings of the AAAI Conference on Human Computation and Crowdsourcing (Vol. 7, pp. 59-67).

Marques-Silva, J., Gerspacher, T., Cooper, M. C., Ignatiev, A., & Narodytska, N. (2021, July). Explanations for Monotonic Classifiers. In International Conference on Machine Learning (pp. 7469-7479). PMLR.

---

### Review · Reviewer_5tob · 2026-03-13

**Summary Of Contributions:**

Disclaimer: I am not an expert when it comes to tabular foundation models, so while I tried to check everything to the best of my knowledge there might be parts that I have missed (especially when it comes to the dynamic developments in the recent ML literature).

The authors propose GAMformer, a transformer that takes a training table as context and, in a single forward pass, predicts the per-feature binned shape functions of a GAM. Predictions are obtained by summing the corresponding feature contributions.

Strengths:

First of all, I find the idea interesting and novel (as already said, I am not an expert for tabular foundation models, so at least from what I can judge the approach seems to be novel), I like the "interpretable tabular foundation model" motivation. Further, I think the model design is reasonable, since binning is a natural choice for non-parametric GAM shape functions, and the bi-attention layer works well with the structure of the tabular data. Empirically, the proposed approach seems to be comparable with GAM baselines.

To sum up, I think the proposal is quite interesting and promising.

Weaknesses:

Frankly speaking, the claim of the paper seems to be fairly larger than the empirical validation. The authors claim to bridge (tabular) foundation models and interpretable machine learning, but the evidence is fairly limited in scope. Relatedly, the pairwise interaction extension is not really an end-to-end foundation model solution, it depends on post-hoc feature-pair selection and cross-validation (so part of the final performance comes from an auxiliary pipeline rather than the core GAMformer idea itself). An other weakness I see is that the paper leans heavily on qualitative shape-function comparisons and average-rank plots, but gives less evidence on whether the learned explanations are stable, faithful or preferable to EBM in an interpretability sense. The authors also acknowledge a scaling limitation, which I find a non trivial limitation for tabular foundation models. It is good though that the authors themselves mention this limitation. Although the method is advertised as amortizing fitting, it is built on a fairly large pretrained transformer and the practical tradeoff versus simply fitting a strong EBM per dataset is not completely clear for me from the presented (empirical) evidence.

Some technical weaknesses (some can be seen as remarks though):

The optimizer description is inconsistent In Section 3.3, the authors say that they train with SGD on synthetic data priors, but Appendix E says the model is trained with AdamW with cosine schedule and specific hyperparameters.

Appendix E says each dataset used during training had a number of feature drawn uniformly between 1 and 10. However, Section 3.4 says that increasing feature dimensions "beyond the 100 used in pretraining" is problematic.

A last inconsistency concerns optimization/regularization hyperparameters. Concretely, Section 3 says the single-forward pass approach eliminates the need for optimization and regularization hyperparameters, yet the higher-order-effects extension uses FAST-based pair selection and chooses a number of pairs by hold-out / cross-validation at inference time.


Minor things:

There are some formatting issues in the Appendix (especially p. 19, 20, where the figures go over the page margins). The authors might want to improve this here.

There are some typos, e.g. "mofification" on p.8, "emmission" on p.5.

The PFN background says that the posterior predictive is approximated "up to an additive constant". That reads not quite right, since for posterior predictive distributions one would normally talk about normalization, not an additive constant.

At some point there is a statement that GAM shape functions are also sometimes called partial dependence plots. Really? That seems wrong. A partial dependence plot is a post hoc marginal effect visualization.

**Audience:**

Yes

**Audience Explanation:**

Since tabular foundation models are a big topic right now I definitely thing that this paper in the intersection to interpretability is interesting for the TMLR audience, and ML community in general.

**Broader Impact Concerns:**

N/A.

**Claims And Evidence:**

No

**Claims Explanation:**

This not a clear no, something in between, because of the reasons mentioned above; I don't think that the empirical validation does the papers claims justice.

**Requested Changes:**

The requested changes largely align with the weaknesses discussed above.

In particular, I would ask the authors to fix the concrete inconsistencies in the training/setup description; clearly separate the base GAMformer model from the pairwise-augmented pipeline used for the strongest benchmark results; better substantiate the interpretability and robustness claims, and clean up conceptual/notational issues and improve the overall presentation.

---

### Decision · Action_Editor_9gkg · 2026-04-25

**Recommendation:** Reject

**Additional Comments:**

The reviewers were unanimous in recommending rejection, and there was no rebuttal from the authors. Based on the reviewer comments as I have summarized above, there is a lot of work to do on this submission.

**Audience:**

No

**Audience Explanation:**

At least two reviewers found the motivation for GAMformer to be unclear. Conventional GAM training is not expensive, so the reviewers questioned whether the substantial cost of pre-training GAMformer is worthwhile compared to fitting a GAM to each new dataset. The apparent limitation to feature dimensions less than 100 (noted by Reviewer 5tob) adds further doubt to this question.

**Claims And Evidence:**

No

**Claims Explanation:**

This submission proposes GAMformer, a combination of tabular foundation models (which are pre-trained on a large number of datasets and predict on a new dataset through in-context learning) with generalized additive models (GAMs) for interpretability.

The overall feeling from the reviews is that the claims are larger than what the empirical validation supports.
- Reviewers commented that the evaluation is more on the qualitative side, with insufficient quantitative comparisons of predictive performance with baselines and no assessment of stability of the shape functions to re-fitting. Reviewer quL9 noted the omission of the NAM baseline.
- The interpretability of the resulting GAMs was also insufficiently supported. Reviewer Rdhp questioned whether the GAMs are indeed interpretable when features are high-dimensional and without smoothness constraints on the shape functions. There was no human evaluation of interpretability.
- Reviewer 5tob noted that the pairwise interaction component is post hoc and does not come from GAMformer itself. The reviewer called for separate evaluation of base GAMformer and GAMformer + pairwise interactions to understand the contributions of each.

Reviewer 5tob also pointed out several inconsistencies in technical details between the main paper and appendix, for example regarding optimizers and training set dimensions.